# Rapid and Sensitive Detection of *Streptococcus iniae* in *Trachinotus ovatus* Based on Multienzyme Isothermal Rapid Amplification

**DOI:** 10.3390/ijms24097733

**Published:** 2023-04-23

**Authors:** Yifen Wang, Jingjing Niu, Minmin Sun, Ziyi Li, Xiangyuan Wang, Yan He, Jie Qi

**Affiliations:** 1Key Laboratory of Tropical Aquatic Germplasm of Hainan Province, Sanya Oceanographic Institute, Ocean University of China, Sanya 572025, China; 2MOE Key Laboratory of Marine Genetics and Breeding, College of Marine Life Sciences, Ocean University of China, 5 Yushan Road, Qingdao 266003, China

**Keywords:** multienzyme isothermal rapid amplification, lateral flow dipsticks, *Streptococcus iniae*, rapid detection

## Abstract

Infectious diseases caused by *Streptococcus iniae* lead to massive death of fish, compose a serious threat to the global aquaculture industry, and constitute a risk to humans who deal with raw fish. In order to realize the early diagnosis of *S. iniae*, and control the outbreak and spread of disease, it is of great significance to establish fast, sensitive, and convenient detection methods for *S. iniae*. In the present study, two methods of real-time MIRA (multienzyme isothermal rapid amplification, MIRA) and MIRA-LFD (combining MIRA with lateral flow dipsticks (LFD)) for the *simA* gene of *S. iniae* were established, which could complete amplification at a constant temperature of 42 °C within 20 min. Real-time MIRA and MIRA-LFD assays showed high sensitivity (97 fg/μL or 7.6 × 10^2^ CFU/mL), which were consistent with the sensitivity of real-time PCR and 10 times higher than that of PCR with strong specificity, repeatability simplicity, and rapidity for *S. iniae* originating from *Trachinotus ovatus*. In summary, real-time MIRA and MIRA-LFD provide effective ways for early diagnosis of *S. iniae* in aquaculture, especially for units in poor conditions.

## 1. Introduction

The *Streptococcus iniae* (*S. iniae*) is a kind of facultative anaerobic Gram-positive pathogen that is beta-hemolytic on sheep blood agar plates [1]. This bacterium can infect a variety of freshwater and marine fish, including *Oreochromis niloticus*, *Paralichthys olivaceus*, *Oncorhynchus mykiss*, *Pagrus pagrus*, and *Trachinotus ovatus* [2,3,4]. It not only destroys the cultural environment, but also causes huge economic losses to the aquaculture industry [3,5]. The researches showed that *simA* is one of the major virulence factors of *S. iniae* and is highly conserved, being the key gene leading to disease outbreaks [6,7]. At the same time, it can also infect humans through wounds and endanger human health [4,8]. Therefore, the early detection of *S. iniae* is increasingly important for prevention and treatment of streptococcosis.

At present, except for conventional physiological and biochemical identification, the detection methods of *S. iniae* mainly include immune-related and molecule-based assays. Immune-related detection methods mainly include enzyme-linked immunosorbent assay (ELISA) [9] and indirect fluorescence antibody technology (IFAT) [10]. Molecular-based detection methods mainly include polymerase chain reaction (PCR) [11] and real-time PCR (RT-PCR) [12]. In recent years, isothermal amplification technology has also been developed, such as Loop-mediated isothermal amplification (LAMP) for the detection of *S. iniae* [5]. Unlike LAMP, PCR, and their derivative techniques, nucleic acid amplification by MIRA technology does not depend on complex temperature variation, and design multiple primer pairs. MIRA technology is the homologous technology of recombinase polymerase amplification (RPA), which is completed in 5–20 min at 37–42 °C, and gets rid of dependence on expensive instruments [13]. The amplification principles of them are the same, but the difference lies in the source and modification of the core proteins. According to the different detection methods, MIRA could be divided into basic-MIRA, real-time MIRA and MIRA-LFD corresponding to agarose gel electrophoresis, real-time fluorescence collection and lateral flow dipsticks, respectively.

In the present study, we developed real-time MIRA and LFD assays to detect *S. iniae*. Through sensitivity, specificity, and repeatability assays, the two methods have strong specificity and good repeatability. Moreover, the sensitivity of them were, consistent with real-time PCR and 10 times higher than PCR which could greatly shorten the reaction time and no longer rely on expensive and complex instruments and equipment. The aim of the investigation was to provide a quick and convenient method for the detection of *S. iniae* in aquaculture farms with poor conditions.

## 2. Results

### 2.1. Evaluation of MIRA Primers

In the primer screening assays, twenty-five primers were amplified by basic MIRA. The gel imaging showed that all the primer combinations amplified the target sequences successfully, and the combination of F4R5 primer yielded the best products on the agarose gel with bright and high amplification efficiency than other combinations (Figure 1). Therefore, F4R5 primer was used for the subsequent assays. The target product amplified by F4R5 was 239 bp in length.

### 2.2. Optimal Reaction Conditions

Since real-time MIRA was able to intuitively present the state of the reaction, it is used to confirm the reaction conditions. The real-time MIRA reaction system was amplified at 37 °C, 39 °C, and 42 °C for 25 min. As shown, the real-time MIRA could be amplified at 37 °C, 39 °C, and 42 °C (Figure 2). The reaction at 42 °C showed the shortest time to reaches the amplification exponential phase and platform stage (20 min) than other groups. Therefore, 42 °C and 20 min were confirmed as the optimum temperature and time, respectively.

### 2.3. Specificity of Real-Time MIRA and MIRA-LFD Assays

The specificity of the real-time MIRA and MIRA-LFD assays showed that only the DNA of *S. iniae* yielded positive results, while the other 10 bacterial samples and the negative control did not show any positive results (Figure 3). The results indicated that real-time MIRA and MIRA-LFD are specific for the detection of *S. iniae*.

### 2.4. Sensitivity of Real-Time MIRA and MIRA-LFD Assays

The diluted DNA (9.7 ng/μL–97 fg/μL) of *S. iniae* were detected by PCR, real-time PCR, real-time MIRA, MIRA-LFD. The results showed the DNA limit of detection was 970 fg/μL for PCR (Figure 4A), and 97 fg/μL for real-time PCR, MIRA-LFD and real-time MIRA (Figure 4B–D). Real-time MIRA, MIRA-LFD, real-time PCR, and PCR assays for bacterial suspension sensitivity were performed using DNA extracted from a dilution series of gradient bacterial suspension (7.6 × 10^7^−10^0^ CFU/mL) of *S. iniae*. The detection limit of bacterial suspension was 7.6 × 10^3^ CFU/mL for PCR (Figure 5A), and 7.6 × 10^2^ CFU/mL for real-time PCR, real-time MIRA, and MIRA-LFD (Figure 5B–D). The above assays showed that real-time MIRA and MIRA-LFD are as sensitive as real-time PCR and 10 times more sensitive than PCR.

### 2.5. Reproducibility of Real-Time MIRA and MIRA-LFD Assays

The reproducibility assays of real-time MIRA and MIRA-LFD were performed using the limit of detection for DNA (97 fg/μL) and bacterial suspension (7.6 × 10^2^ CFU/mL) as templates. The results showed that the exponential phase of real-time MIRA and the color of MIRA-LFD are basically the consistent in triplicate (Figure 6A–D).

### 2.6. Practicability of Real-Time MIRA and MIRA-LFD Assays

The livers of 12 golden pompano were selected to evaluate the practicability of *S. iniae* in real-time MIRA and MIRA-LFD with real-time PCR as the standard. The results showed that five samples (No. 2, 3, 5, 10, 11) were positive detected by real-time MIRA and MIRA-LFD (Figure 7A,B), which consistent with that of real-time PCR (Figure 7C). The results indicated that real-time MIRA and MIRA-LFD were effective in detection the *S. iniae*.

## 3. Discussion

*S. iniae* can infect a wide range of fish species, including many economically related aquaculture species. It is estimated that the global aquaculture industry loses billions of dollars annually due to *S. iniae* [14]. Currently, in addition to using antibiotics to treat streptococcal disease, vaccines are also targeted research, but it is still difficult to obtain effective vaccines. The abuse of antibiotics not only increases drug resistance and destroys the stability of water, but also causes harm such as antibiotic residues. Therefore, it is necessary to detect and prevent the infection of *S. iniae* as early as possible. Traditional detection methods, such as physiological and biochemical identification, PCR, and ELISA, cannot meet the requirements of grass-roots detection, nor can they be operated conveniently. Physiological and biochemical identification and ELISA usually take 2–3 days or more [15]. Although the identification time of PCR is greatly shortened [16], it needs to rely on expensive instruments and equipment. Even though, the LAMP technology developed in recent years still needs 1.5–2 h to complete the whole detection process [17,18]. MIRA technology, also known as the homologous technology of RPA, can not only amplify at constant temperature without designing primers for multiple regions, but also greatly improve the amplification efficiency and provide a new idea for POCT [19]. With the development of thermostatic amplification technology, some companies have also developed portable thermostatic amplification instruments that can be used for real-time fluorescence collection for diagnosis [20]. LFD can directly judge the diagnosis result through the naked eye [21,22]. Therefore, we reported the development and validation of real-time MIRA and MIRA-LFD assays, which were used to detect *S. iniae* in liver samples of golden pompano. The amplification process from both MIRA takes about 20 min or less. Again, they showed well sensitivity as real-time PCR, ten times more than PCR, and rapidity and convenient, less equipment requirement, especially for the grass-roots level such as aquaculture farms with limited resources [23,24,25,26,27].

The technology of molecular diagnosis includes three steps: sample preparation, amplification, and detection. In addition, sample preparation was considered to be the greatest challenge [28]. In this study, two methods were used to extract DNA from *S. iniae*. One is the DNA Kit, and the other is lysis of the sample using the nucleic acid releaser. The DNA kits needed to spend a long-time extracting DNA, usually about 1 h, and rely on precise laboratory instrumentation, which limit its application in rapid and low-resource diagnosis. Therefore, this method of the nucleic acid releaser was adopted to extract DNA, which only needs to be split at 95 °C for 5 min. Through PCR, real-time PCR, real-time MIRA, and MIRA-LFD detection, it was found that the DNA extracted by the nucleic acid releaser accelerated the reaction process, which provided a good means for sample preparation.

In order to detect pathogenic bacteria, many detection methods have been established. PCR is undoubtedly widely used in the detection of pathogenic bacteria, including *Staphylococcus aureus* [29], *salmonella* [30], *Vibrio cholerae* [31], *Acinetobacter baumannii* [24], *Phytophthora sojae* [32], and *Pseudomonas aeruginosa* [33]. With the continuous development of molecular diagnosis, many thermostatic amplification techniques such as LAMP, NASBA [34], and HDA [35] are gradually applied to the diagnosis of pathogenic bacteria. Among these, LAMP technology, which has been widely studied, has also been applied to the detection of *S. iniae*. In previous studies, the sensitivity of PCR to detect *S. iniae* was per reaction mixture and 25 pg of DNA [11], while that of LAMP was 100 fg [36]. In addition, the detection limit of the bifunctional probes-labeled AuNPs combined with LAMP for detection of *S. iniae* is 10^2^ CFU [5]. The sensitivity of the real-time MIRA and MIRA-LFD methods established in this study to detect *S. iniae* is 97 fg/μL and 7.6 × 10^2^ CFU/mL, which is far higher than the sensitivity of PCR, and basically consistent with the sensitivity of LAMP. The MIRA technique does not require a long amplification time and primer design based on the six regions of the target gene as LAMP does.

We all know that detection by PCR requires the aid of agarose gel electrophoresis to distinguish amplicons, and the aid of gel imager to visualize the final results; real-time PCR achieves real-time detection of the entire amplification process through fluorescence signal accumulation, making it easy to analyze the entire amplification process. For MIRA technology, there are currently three detection methods: basic, exo (real-time MIRA), and nfo (MIRA-LFD). Because basic MIRA requires gel imaging, the process is cumbersome and relies on gel dispenser and gel imager, so it is not suitable for POCT. Therefore, this study mainly establishes two detection methods of exo and nfo, which do not rely on professional operators and instruments to decrease the time for the detection of *S. iniae*. The PCR method established by Mata et al. [11] takes 3 h. While the LAMP technology established by Han et al. [36] and Cai et al. [18] spent the detection time within 90 min, but in this study, real-time MIRA and MIRA-LFD were able to be completed in 30–40 min, significantly reducing the detection time.

In conclusion, real-time MIRA and MIRA-LFD assays were sensitive, specific, and convenient for the detection of *S. iniae*. Meanwhile, the real-time MIRA and MIRA-LFD were used to detect *S. iniae* in a shorter time, requiring fewer instruments and equipment, and easier to operate. It is very essential for the rapid detection of *S. iniae* in aquaculture farms, especially for that with low-resource units.

## 4. Materials and Methods

### 4.1. Fish and Bacterial Strains

All fish were obtained from Guangxi Vim Marine Technology Co., Ltd (Beihai, China). A total of 11 strains (one *S. iniae*, and ten non-*S. iniae*) were used to analyze the specificity of the real-time MIRA and MIRA-LFD assays, including *Vibrio campbellii*, *Vibrio alginolyticµs*, *Vibrio harveyi*, *Vibrio Parahemolyticµs*, *Vibrio rotiferianµs*, *Staphylococcµs epidermidis*, *Staphylococcµs aµreµs*, *Klebsiella pneµmoniae*, *Lactococcµs garvieae*, *Streptococcµs dysgalactiae*, and *Streptococcµs iniae*. All bacteria were stored in our lab.

### 4.2. Genomic DNA Extraction

Two methods were used to extract the genomic DNA of *S. iniae*. One was extracted by a Tiangen Bacterial DNA Kit (Tangent Biotech Co., Ltd., Beijing, China) only for the DNA sensitivity assay, according to the manufacturer’s instructions. The extracted DNA concentration was measured with Qubit 4.0 and stored at −20 °C; another one was added to a nucleic acid releaser (GenDx, Suzhou, China) at 95 °C for 5 min to release genomic DNA, for assays other than DNA sensitivity assay.

### 4.3. Design of Primers and Probes

Twenty-five primers for *simA* gene (GenBank accession number: JF330100.1) were designed with Primer Premier 5.0 for MIRA assay, then the optimal primers were determined by 1.5% agarose gel. The information on primers is listed in Table 1. The probes of real-time MIRA and MIRA-LFD were designed according to the sequence between the optimal primers, and corresponding markers were added. The primers and probes for real-time MIRA and MIRA-LFD are listed in Table 2.

### 4.4. Establishment of MIRA, Real-Time MIRA, and MIRA-LFD Assays

The MIRA was initiated by using the MIRA basic kit (Amp-Future, Weifang, China), and the reaction was performed in a 25 μL volume containing 1 μL of forward primer (10 µM), 1 μL of reverse primer (10 µM), 14.7 μL of A buffer, 5.05 μL of nuclease-free water and 2 μL of sample DNA, and then the reaction was started by adding 1.25 μL of B buffer (280 mM). The products of MIRA performed at 42 °C for 30 min were electrophoresed through agarose gels (1.5%, *w*/*v*) at 120 V for 20 min and detected by Invitrogen iBright CL750 (Thermo Fisher Scientific, Waltham, MA, USA).

The real-time MIRA was initiated by using the MIRA exo kit (Amp-Future, China), and the reaction was performed in a 25 μL volume containing 1 μL of forward primer (10 µM), 1 μL of reverse primer (10 µM), 14.7 μL of A buffer, 0.5 μL of probes (10 µM), 4.55 μL of nuclease-free water and 2 μL of sample DNA, and then started the reaction by adding 1.25 μL of B buffer (280 mM). All reactions of real-time MIRA were conducted with Bio-Rad CFX96™ Touch™ Real-time PCR Detection System (Bio-Rad, Hercules, CA, USA). To determine the optimal amplification temperature, real-time MIRA was performed at 37 °C, 39 °C, and 42 °C for 25 min. The optimum reaction time is determined based on the amplification curve of the determined optimum reaction temperature.

The MIRA-LFD was initiated by using the MIRA nfo kit (Amp-Future, Weifang, China), and the reaction system was fellow as the real-time MIRA. The product of MIRA-LFD is detected with lateral flow dipstick (JY0201, Baoying Tonghui Biotechnology Co., Ltd., Beijing, China) and the result are read within 5–10 min. If both the detection line and control line display bands, the result is positive; if only the control line displays a band, the result is negative.

### 4.5. Specificity of Real-Time MIRA and MIRA-LFD Assays

The specificity of real-time MIRA and MIRA-LFD assays was performed using the DNA of *S. iniae*, *V. campbellii*, *V. alginolyticµs*, *V. harveyi*, *V. Parahemolyticµs*, *V. rotiferianµs*, *S. epidermidis*, *S. aµreµs*, *K. pneµmoniae*, *L. garvieae*, *S. dysgalactiae,* and nuclease-free water (negative control) as templates. The results of real-time MIRA analyzed with GraphPad Prism 8.0.1.

### 4.6. Sensitivity of Real-Time MIRA and MIRA-LFD Assays

The sensitivity of real-time MIRA and MIRA-LFD assays included two aspects: DNA sensitivity and bacterial suspensions sensitivity.

To evaluate the DNA sensitivity, genomic DNA of *S. iniae* was extracted and diluted a range of 9.7 ng/μL to 9.7 fg/μL. Bacterial suspensions sensitivity was performed by extracting DNA from diluted bacterial suspensions of *S. iniae* in the range of 7.6 × 10^7^–10^0^ CFU/mL. All DNA were used for amplification of PCR, real-time PCR, real-time MIRA and MIRA-LFD. The condition and system of the real-time PCR reaction referred to Yolanda Torres-Corral and Ysabel Santos [12]. The PCR reaction was performed in a 25 μL volumes containing 12.5 μL of 2 × Taq Master Mix (Vazyme, Nanjing, China), 1 μL of forward primer (10 µM), 1 μL of reverse primer (10 µM), 8.5 μL of nuclease-free water, and 2 μL of template DNA. After a denaturation step of 95 °C for 5 min, 35 serial cycles of a denaturation step of 95 °C for 30 s, annealing at 50 °C for 30 s and extension at 72 °C for 30 s were performed, follow by a final extension step of 72 °C for 5 min. The information of primers for PCR and real-time PCR was showed in Table 3.

### 4.7. Reproducibility of Real-Time MIRA and MIRA-LFD Assays

The limits of detection determined by the sensitivity assays of DNA and bacterial suspensions were used as templates for triplicate assays to verify the reproducibility of real-time MIRA and MIRA-LFD.

### 4.8. Practicability of Real-Time MIRA and MIRA-LFD Assays

To evaluate the efficacy of real-time MIRA and MIRA-LFD assays in actual detection, livers of golden pompano (*Trachinotus ovatus*) suspected to be infected with *S. iniae* obtained from Guangxi Vim Marine Technology Co., Ltd. were selected to extract DNA for real-time PCR, real-time MIRA, and MIRA-LFD.

## 5. Conclusions

Real-time MIRA and MIRA-LFD assays for rapid detection of *S. iniae* was developed and established, which could reach detection limits of 97 fg/μL and 7.6 × 10^2^ CFU/mL within 20 min at 42 °C and show high specificity and repeatability. The evaluation effect of detection for actual sample is consistent with that of real-time PCR. The real-time MIRA and MIRA-LFD established in this study provide effective means for rapid diagnosis of *S. iniae* in the aquaculture.

## 6. Patents

The work reported in this manuscript may generate patents in the future, but it is not yet available.

## Figures and Tables

**Figure 1 ijms-24-07733-f001:**
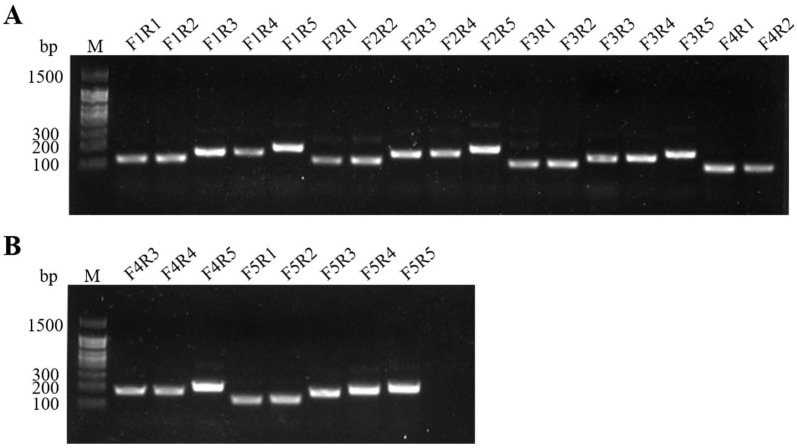
Amplicons of twenty-five primer combinations using the basic MIRA assay. F: forward primer; R: reverse primer; M: 100 bp DNA Ladder. (**A**) The primer combinations of F1R1–F4R2; (**B**) The primer combinations of F4R3–F5R5.

**Figure 2 ijms-24-07733-f002:**
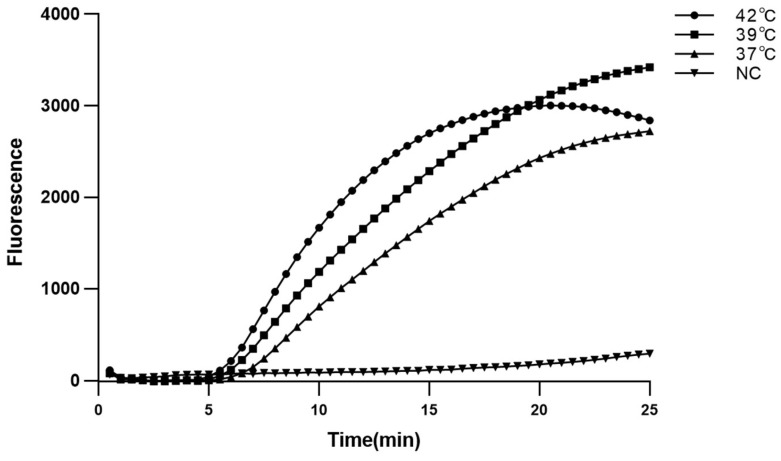
Optimum reaction temperature for real-time MIRA assays. NC: negative control.

**Figure 3 ijms-24-07733-f003:**
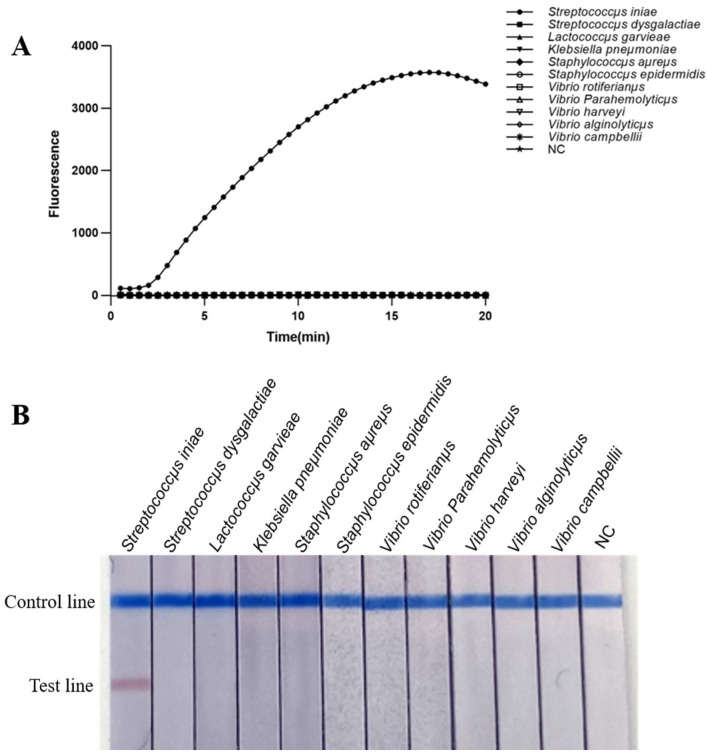
Specificity of real-time MIRA and MIRA-LFD assays. NC: negative control. (**A**) Specificity of real-time MIRA assays; (**B**) Specificity of MIRA-LFD assays.

**Figure 4 ijms-24-07733-f004:**
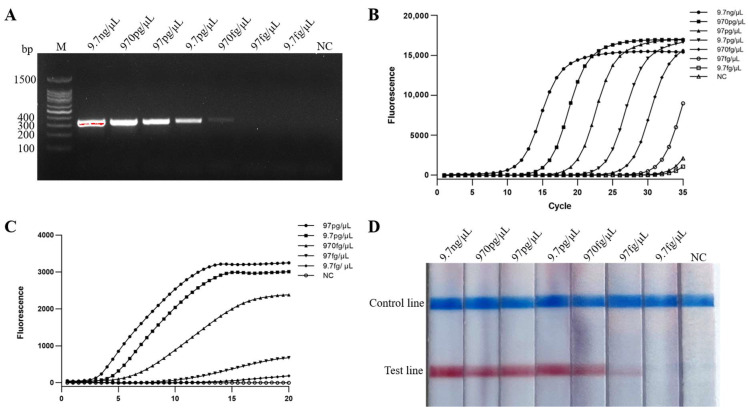
Comparison of DNA sensitivity. NC: negative control. DNA detection limits were determined using DNA diluted at 9.7 ng/μL–9.7 fg/μL as template for PCR (**A**), real-time PCR (**B**), real-time MIRA (**C**), and MIRA-LFD (**D**) assays.

**Figure 5 ijms-24-07733-f005:**
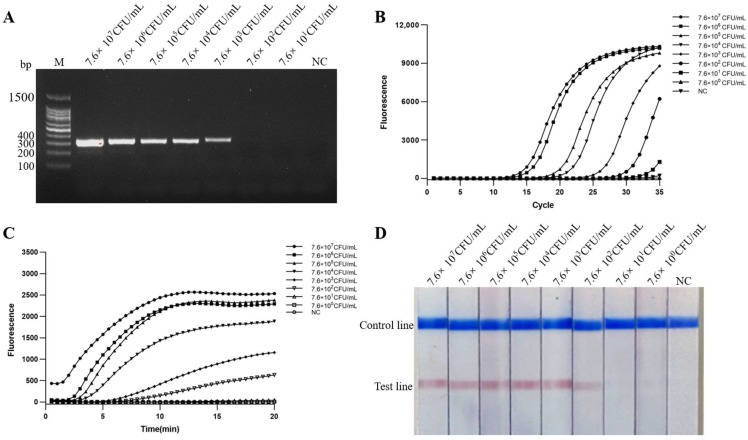
Comparison of the sensitivity of bacterial suspensions. NC: negative control. The detection limits of bacterial suspensions were determined using DNA extracted from a dilution series of gradient bacterial suspension samples containing 7.6 × 10^7^−10^0^ CFU/mL of *S. iniae* as templates for PCR (**A**), real-time PCR (**B**), real-time MIRA (**C**), and MIRA-LFD (**D**) assays.

**Figure 6 ijms-24-07733-f006:**
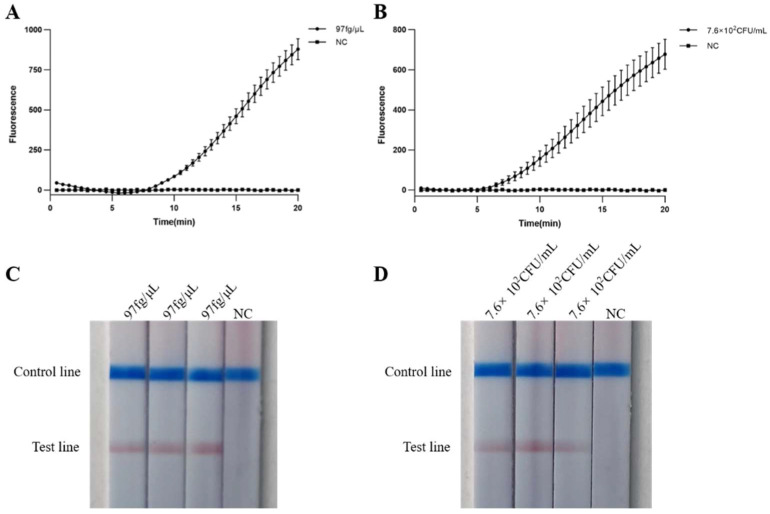
Reproducibility of real-time MIRA and MIRA-LFD assays. NC: negative control. (**A**,**B**) DNA detection limit (97 fg/μL) and bacterial suspensions detection limit (7.6 × 10^2^ CFU/mL) of the real-time MIRA reproducibility assays, respectively. (**C**,**D**) DNA minimum detection limit (97 fg/μL) and bacterial fluid minimum detection limit (7.6 × 10^2^ CFU/mL) of the MIRA-LFD reproducibility assays, respectively.

**Figure 7 ijms-24-07733-f007:**
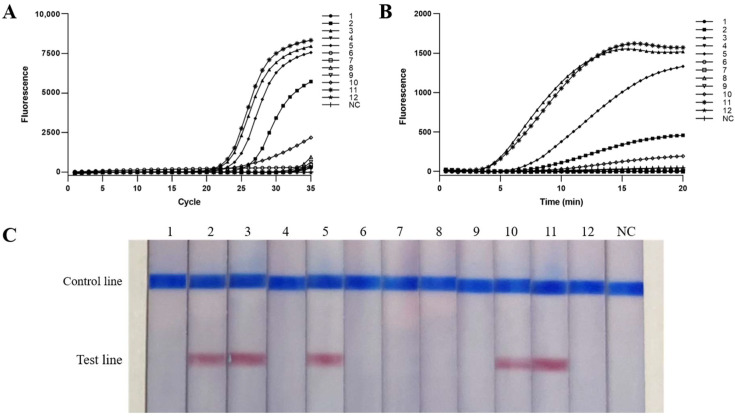
Evaluation of the practicability of real-time MIRA and MIRA-LFD based on real-time PCR. 1–12:12 samples of suspected *S. iniae* infection; NC: negative control. (**A**–**C**) The detection results of real-time PCR, real-time MIRA, and MIRA-LFD respectively.

**Table 1 ijms-24-07733-t001:** The primers of *simA* for MIRA assay. Location according to JF330100.1.

Name	Sequence (5′–3′)	Location
*simA*-F1	TAGAAGCGGCTAAGAAAGAAGCAGAAGAA	923–951
*simA*-F2	TGACTAAAGCATTAGAAGCGGCTAAGAAA	911–939
*simA*-F3	ACTAAAGCATTAGAAGCGGCTAAGAAAGA	913–941
*simA*-F4	TAAAGCATTAGAAGCGGCTAAGAAAGAAG	915–943
*simA*-F5	AAGCATTAGAAGCGGCTAAGAAAGAAGCAG	917–946
*simA*-R1	GTGCTTTCTCAAGGTCTTTTTCAAGTTCT	1057–1029
*simA*-R2	CTGCTTGTGCTTTCTCAAGGTCTTTTTCAA	1063–1034
*simA*-R3	TCCAATTCAGCTTTTGTTTCTGCTAGTTTA	1112–1083
*simA*-R4	TTTCCAATTCAGCTTTTGTTTCTGCTAGTT	1114–1085
*simA*-R5	CAATAGTTGCTTCAAGTTCTGCTTTTTCA	1153–1125

**Table 2 ijms-24-07733-t002:** Primers and probes for real-time MIRA and MIRA-LFD.

Assay	Name	Sequence (5′–3′)
Real-time MIRA	exo-F	TAAAGCATTAGAAGCGGCTAAGAAAGAAG
	exo-R	CAATAGTTGCTTCAAGTTCTGCTTTTTCA
	exo-P	TTTCCAATTCAGCTTTGTTTCTGCTAGT(i6FAMdT) (idSp)(iBHQ1dT)AGTTTCAAGGTCTTTAG
MIRA-LFD	nfo-F	TAAAGCATTAGAAGCGGCTAAGAAAGAAG
	nfo-R	(Biotin)CAATAGTTGCTTCAAGTTCTGCTTTTTCA
	nfo-P	(FAM)TTTCCAATTCAGCTTGTTTCTGCTAGTT (idSp)TAGTTTCAAGGTCTTTAG(C3 Spacer)

**Table 3 ijms-24-07733-t003:** Primers for PCR and real-time PCR (*SilldP*-F and *SilldP*-R).

Name	Sequence (5′–3′)
PCR-F	TGAAGAGCTTGACAAACTAAATG
PCR-R	ACTTGCTGTGAAGAATGGGTTA
SilldP-F	ACACAGGTGAGCACGCTAAA
SilldP-R	CGTCACCATCGTCTTGGTCA

## Data Availability

Not applicable.

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
