# Peer review of "Rapid and Sensitive Detection of Streptococcus iniae in Trachinotus ovatus Based on Multienzyme Isothermal Rapid Amplification"

_ijms, 2023, doi:10.3390/ijms24097733_

Round 1

Reviewer 1 Report

The paper is well written and research well designed.

Reviewer 3 Report

 The manuscript reported the research results of rapid and sensitive detection of Streptococcus iniae in Trachinotus ovatus based on multienzyme isothermal rapid amplification in detail. However, some major issues should be mentioned.

 (1)   Multienzyme isothermal rapid amplification is a homologous technology of recombinase polymerase amplification (RPA), while the reference 11&17 are not well cited, and the difference between the two methods was not explained in detail.

(2)   As a developing method in rapid diagnosis, RPA and MIRA are extensively studied. The main differences of the studies are the primers and detection targets. The novelty of the manuscript is pool and may not be interested to the readers to meet the high quality publishing on IJMS.

      Such as Rapid detection of Staphylococcus aureus using a novel multienzyme isothermal rapid amplification technique. Front. Microbiol. 2022, 13:1027785; Development of a multienzyme isothermal rapid amplification and lateral flow dipstick combination assay for bovine coronavirus detection.2023, Front. Vet. Sci. 9:1059934. Development and evaluation of a rapid and sensitive multienzyme isothermal rapid amplification with a lateral flow dipstick assay for detection of Acinetobacter baumannii in spiked blood specimens. Front. Cell. Infect, 2022, Microbiol. 12:1010201.

(3)   In the conclusion section, the manuscript reported that two methods were established in for rapid diagnosis of S. iniae, to be more more precisely, development and validation of the MIRA-LFD assay for detection of S. iniae can be addressed, or development and validation of the primers used for detection of S. iniae,other than established two methods.

Round 2

Reviewer 3 Report

The author provided some in-depth explanations, and the manuscript can be accepted in present form.